# Neuroprotective Effects and Therapeutic Potential of Transcorneal Electrical Stimulation for Depression

**DOI:** 10.3390/cells10092492

**Published:** 2021-09-21

**Authors:** Wing-Shan Yu, So-Hyun Kwon, Stephen Kugbere Agadagba, Leanne-Lai-Hang Chan, Kah-Hui Wong, Lee-Wei Lim

**Affiliations:** 1Neuromodulation Laboratory, School of Biomedical Sciences, Li Ka Shing Faculty of Medicine, The University of Hong Kong, Hong Kong, China; yuwsw@connect.hku.hk (W.-S.Y.); u3544616@connect.hku.hk (S.-H.K.); wkahhui@um.edu.my (K.-H.W.); 2Department of Electrical Engineering, City University of Hong Kong, Hong Kong, China; sagadagba3-c@my.cityu.edu.hk (S.K.A.); leanne.chan@cityu.edu.hk (L.-L.-H.C.); 3Department of Anatomy, Faculty of Medicine, Universiti Malaya, Kuala Lumpur 50603, Malaysia

**Keywords:** TES, transcorneal electrical stimulation, neuromodulation, depression, antidepressant, neuroprotection

## Abstract

Transcorneal electrical stimulation (TES) has emerged as a non-invasive neuromodulation approach that exerts neuroprotection via diverse mechanisms, including neurotrophic, neuroplastic, anti-inflammatory, anti-apoptotic, anti-glutamatergic, and vasodilation mechanisms. Although current studies of TES have mainly focused on its applications in ophthalmology, several lines of evidence point towards its putative use in treating depression. Apart from stimulating visual-related structures and promoting visual restoration, TES has also been shown to activate brain regions that are involved in mood alterations and can induce antidepressant-like behaviour in animals. The beneficial effects of TES in depression were further supported by its shared mechanisms with FDA-approved antidepressant treatments, including its neuroprotective properties against apoptosis and inflammation, and its ability to enhance the neurotrophic expression. This article critically reviews the current findings on the neuroprotective effects of TES and provides evidence to support our hypothesis that TES possesses antidepressant effects.

## 1. Introduction

Major depressive disorder, commonly known as depression, is the leading cause of disability worldwide [1]. It is considered a major global disease burden, with more than 4.4% of the world’s population estimated to suffer from depression, and 800,000 depression-related suicide cases annually [2]. The economic cost of depression in US adults exceeded USD 300 billion in 2018 [3]. According to the Diagnostic and Statistical Manual of Mental Disorders 5th Edition (DSM-V), depression is primarily characterized by anhedonia and sadness persisting for at least 2 weeks, which is accompanied by secondary symptoms such as fatigue, sense of worthlessness, psychomotor agitation, changes in appetite or weight, sleep difficulties, loss of concentration, and/or recurrent thoughts of suicide [4]. The burden of depression is further compounded by the high comorbidity of physical disorders [5], as evidenced by its close association with pain [6,7], dementia [8], type 2 diabetes [9,10], cardiovascular diseases [11,12,13,14], and cancers [15]. Depression not only leads to additional medical and financial costs, but also aggravates the prognosis or even mortality in diseased populations.

Depression is commonly treated with psychotherapy and medications that generally target the reuptake of neurotransmitters [16]. Nevertheless, up to 60% of patients inadequately respond to drug treatments, in which approximately 10–30% of patients develop treatment-resistant depression (TRD) with failure to respond to two or more types of antidepressant treatments [17,18,19,20]. Electroconvulsive therapy (ECT) is considered a common treatment for TRD and the remission rate is reported at approximately 50%, with nearly half of the patients relapsing within the first year following ECT [21,22,23,24,25]. Given the increasing prevalence of depression and unsatisfactory outcomes of currently available treatments, there is an urgent need to develop alternative therapeutic options for the treatment of depression.

Neuromodulation is a technology which utilizes electrical stimulation to modulate the nervous system functioning [26]. It is emerging as a promising therapeutic approach against various psychiatric and neurological disorders [20]. Among the different types of neuromodulation-based techniques, transcorneal electrical stimulation (TES) is a non-invasive treatment that is reported to improve visual functions in various ophthalmological conditions [27]. Although current studies on TES mostly focus on its use in ophthalmology, TES is also demonstrated to induce neurobehavioural changes including antidepressant-like behaviour in corneally kindled models [28]. Surprisingly, apart from activating the retina and associated downstream visual structures, enhanced activities are additionally reported in the prefrontal cortex (PFC) and parahippocampal gyrus (PHG) following TES application [29,30,31]. Although the modulation effects of TES in these non-visual brain regions have yet to be confirmed, both PFC and PHG are involved in mood alterations [32], suggesting that TES may have a role in regulating emotion. The potential neuroprotective properties of TES are possibly achieved through regulating neuroplasticity [30,33], neurotrophic expression [34,35,36,37,38,39,40], inflammatory responses [39,41,42,43,44], apoptosis [36,45,46,47], glutamate metabolism [48], and retinal blood flow [49,50]. The putative neuroprotective effects of TES on mood control are further supported by its shared mechanisms of action with current antidepressant treatments, including its neuroprotective effects against apoptosis and inflammation, as well as its ability to promote neurotrophic expression. This review aims to discuss the neuromodulation potential of TES as a treatment for depressive disorders and the neuroprotective mechanisms of action that might contribute to the antidepressant-like responses.

## 2. Connections between Visual and Emotional Systems

A visual–emotional connectivity can be established through cortical and subcortical routes, as visual signals are directed to brain regions that are also involved in emotional regulation (Figure 1) [51,52,53,54,55,56,57]. In the cortical pathway, visual outputs generated in the retina are first transmitted to the occipital visual cortex via the lateral geniculate nucleus of the thalamus. From the visual cortex, the information is either projected dorsally to the posterior parietal cortex, or ventrally to the inferior temporal cortex, from which they interconnect with the PFC and limbic structures, including the amygdala and hippocampus [51,54,55,56,57]. Alternatively, visual information can be conveyed to the thalamic pulvinar through the superior colliculus as a subcortical sensory input to the amygdala [52,53,54]. Direct and indirect functional interactions among the amygdala, hippocampus, and PFC significantly contribute to the regulation of emotions [56,58]. Given the close functional link between the visual and emotional brain regions, it is unsurprising that growing evidence supports the bidirectional association between visual impairment and depression. It has been reported that ophthalmic patients with various eye diseases have a 25% overall prevalence of depressive symptoms, which was significantly higher than that in healthy controls [59]. Likewise, a notable increase in the risk of depression in both blind and visually impaired individuals was identified in a national longitudinal study conducted in South Korea [60]. Conversely, depression may also have a negative impact on visual function. Among retinitis pigmentosa (RP) patients, individuals with depressive symptoms had markedly worse subjective visual function and vision-related quality of life irrespective of the severity of the RP disease [61]. These findings were further supported by a national representative study in the US, which found a longitudinal and bidirectional association between visual impairment and symptoms of mental illness [62]. It was reported that patients with impaired vision were 33% more likely to develop depression, whereas those with depressive symptoms had a 37% increased risk of reporting vision loss in the future. Given the close association between depression and visual impairment, treatments that were intended for ophthalmology disorders may well be effective in ameliorating emotional conditions, and vice versa.

## 3. Application of TES in Ophthalmology

Transcorneal electrical stimulation is a non-invasive neuromodulation technique developed as a therapy for various ophthalmological diseases. The application of TES is practical and easy to administer. Contact-lens-like electrodes are placed on the corneas of subjects and connected to a stimulator and stimulus isolator unit, while an inactive reference electrode is connected to the skin surrounding the eye [27]. This has been shown to have beneficial effects on the retina and optic nerve, leading to visual improvements in both preclinical [34,35,36,42,44,48,63,64,65,66,67,68,69,70] and clinical settings [71,72,73,74,75,76].

### 3.1. Preclinical Studies

Morimoto et al. were the first to report the use of TES in enhancing the survival of retinal ganglion cells (RGCs) after optic nerve transection in rats [34]. The neuroprotective effects of TES on RGCs were found to be associated with the upregulation of insulin growth factor 1 (IGF-1) by Müller glial cells in the retina [34,35]. The research team later demonstrated that the repeated administration of TES at optimal stimulation parameters (pulse width: 1 or 2 ms/phase; stimulation amplitude: 100 or 200 μA; stimulation frequency: 1, 5, or 20 Hz; stimulation duration: >30 min) after optic nerve injury in rats provided the best neuroprotection and led to the regeneration of retinal axons [35,63]. Concordantly, the neuroprotective effects of TES on RGCs were also shown in rodent models of optic nerve crush [64,65,66], non-arteritic ischaemic optic neuropathy [67], ischaemic retinas [48], acute ocular hypertensive injury [42], and glaucoma [44]. The therapeutic applications of TES were not limited to preserving RGCs and their axons. In various transgenic models of retinal degeneration mimicking the pathological changes in human RP, TES treatment was reported to delay the degeneration of photoreceptors and ameliorate retinal functions [68,69,70]. These findings are in line with the findings of Ni et al., which demonstrated that TES could rescue photoreceptors and preserve electroretinogram (ERG) responses after excessive intense light exposure [36].

### 3.2. Clinical Studies

The therapeutic effects of TES in ophthalmology were further investigated in human subjects. In a study of 24 RP patients, six sessions of weekly TES, at a strength of 150% of the individual’s electrical phosphene threshold (EPT), significantly increased the visual field area and ERG scotopic b-wave amplitude [71]. However, these results were not reproducible in a follow-up study on 52 RP patients administered for 52 consecutive weeks with TES, possibly due to the natural progression of the disease [72]. Nevertheless, there was a significant improvement in the photopic ERG responses, suggesting a positive effect of TES on cone functions. In a similar study involving 13 patients with retinal artery occlusion, six sessions of TES at 150% EPT improved photoreceptor activity as measured by an increased scotopic a-wave slope [73]. Two other small-scale clinical trials further reported that TES treatment had positive outcomes on retinal artery occlusion, with improvements observed in multifocal ERG, visual field, and visual acuity [74,75]. Another study provided further evidence of the therapeutic effects of TES, which showed acuity enhancements in two of three patients with non-arteritic ischaemic optic neuropathy and four of five patients with traumatic optic neuropathy [76].

## 4. Mechanisms of Action of TES

Although the therapeutic efficacy of TES in visual-related diseases was investigated in various preclinical and clinical studies, the precise mechanisms of its neuroprotective effects remain obscure. Several mechanisms were proposed to underlie the remedial effects of TES, including neurotrophic, neuroplastic, anti-inflammatory, anti-apoptotic, anti-glutamatergic, and vasodilation mechanisms (Figure 2).

### 4.1. Neurotrophic Mechanism

The neurotrophic mechanism is one of the prevailing theories that attributes the effects of TES to its ability to upregulate neurotrophin expression. Neurotrophins are a family of regulatory factors that are essential for the growth, survival, and functions of neurons [77]. In rodent models of optic nerve injury, TES effectively upregulated IGF-1 in Müller cells, and promoted RGC survival and axonal regeneration [34,35]. Interestingly, the regeneration of RGC axons was inhibited in the presence of an IGF-1 receptor antagonist, indicating the important role of the TES-induced production of IGF-1 in neuroprotection. Concordantly, TES positively modulated the expression of the brain-derived neurotrophic factor (BDNF) and ciliary neurotrophic factor (CNTF) in the retina, which prolonged the survival of photoreceptors and preserved retinal functions in mouse models of photoreceptor degeneration induced by light or N-methyl-N-nitrosourea [36,46]. Studies using Müller cell cultures also demonstrated that TES could stimulate the production of IGF-1, BDNF, and CNTF, resulting in significant induction of Müller cell proliferation [37,38,39,40]. The induction of these growth factors by electrical stimulation was suggested to be dependent on the activation of L-type voltage-dependent calcium channels (L-VDCCs), as blocking L-VDCCs downregulated IGF-1 and BDNF, as well as attenuating Müller cell proliferation [37,38,40].

### 4.2. Neuroplasticity Mechanism

The neuroplasticity mechanism of TES was demonstrated in electroencephalogram (EEG) studies. It was reported that TES in healthy rats led to EEG after-effects in the visual cortex, as demonstrated by a remarkable increase in theta power that lasted beyond the stimulation period by 15 min [33]. Likewise, TES in a mouse model of retinal degeneration increased the EEG power bands in both the visual and prefrontal cortices of awake animals [30]. In both EEG recording sites, the delta oscillations dominated over other frequency bands and were maintained for up to 3 weeks after stimulation. Given the association between slow delta waves and neuroplasticity, these results further reflected the neuroplastic after-effects of TES. Together, these studies suggested that TES could affect neurophysiological activity beyond the stimulation period, contributing to long-term depression or potentiation. The findings in rodent models paralleled the findings from a human study of optic nerve lesions that showed an increased EEG alpha spectrum in the occipital sites after electrical stimulation [78]. The author proposed that TES mediated visual functions through the induction of synaptic plasticity and neuronal synchronization in the occipital region. Moreover, TES was shown to activate the occipital visual cortex by stimulating RGCs, which generated action potentials that transmitted through the retinogeniculate pathway to the visual cortex [79]. When RGC activity was inhibited by tetrodotoxin, a sodium ion channel blocker, the generation of electrically evoked potentials was abolished in the visual cortex. This further supported the idea that TES conferred its effects through enforcing excitation potentials and synchronizing neuronal activity in the visual pathway. The repeated activation of the visual system could strengthen synaptic function, facilitating the transfer and processing of visual information, leading to visual improvements.

### 4.3. Anti-Inflammatory Mechanism

The anti-inflammatory mechanism suggested that the neuroprotective effects of TES can be explained through the blockade of microglial activation. In an in vitro model of light-induced photoreceptor degeneration, electrical stimulation was shown to suppress the activation of microglia, which in turn inhibited the secretion of pro-inflammatory interleukin (IL)-1β and the tumour necrosis factor (TNF)-α, leading to decreased photoreceptor cell death [39]. These results were consistent with rodent experiments that showed that TES had immunomodulatory effects in suppressing retinal microglial activation after acute ocular hypertensive injury [42], optic nerve transection [43], and in a glaucoma model [44]. The improvement in RGC survival was accompanied by the upregulation of the anti-inflammatory cytokine IL-10 and the reduction of pro-inflammatory factors IL-6, TNF-α, and cyclooxygenase-2 [42,43]. The activation of nuclear factor κB, a central mediator of inflammation, was also suppressed by TES in the retina [42]. Together, these findings provided evidence for the neuroprotective effect of TES on inflammation through inhibiting microglial activation and mediating the secretion of inflammatory cytokines.

### 4.4. Anti-Apoptosis Mechanism

Another candidate mechanism of the neuroprotective role of TES is through its anti-apoptotic effects. It was reported that TES could activate the intrinsic survival system by increasing the gene and protein expression of anti-apoptotic B-cell lymphoma 2 (Bcl-2) and by decreasing pro-apoptotic Bcl-2-associated X protein (Bax) in the retina, which protected photoreceptors from light-induced degeneration [36]. In further support of the anti-apoptotic effects of TES, a microarray analysis revealed that TES induced the differential expression of 490 genes, including the downregulation of pro-apoptotic Bax and the TNF Receptor Superfamily Member 12A [45]. Additionally, calpain-2, a calcium-sensitive protease that executes neuronal apoptosis, was also found to be downregulated after TES treatment, suggesting that the electrical stimulation could alleviate pro-apoptotic cellular calcium overload [46,47]. These findings showed that TES had a regulatory role in the apoptotic cascade, thereby preventing neurons from programmed cell death.

### 4.5. Anti-Glutamatergic Mechanism

The ability to alleviate glutamate-mediated excitotoxicity is another putative mechanism responsible for the therapeutic effects of TES. The excessive release of the excitatory neurotransmitter glutamate by damaged neurons could lead to further neuronal dysfunction and death as a result of calcium overload and oxidative stress, a process called glutamate excitotoxicity [80]. A study by Wang et al. reported that TES treatment in a rat model of retinal ischaemia resulted in the elevated expression of glutamine synthetase in Müller cells, subsequently protecting RGCs [48]. The induction of glutamine synthetase is an essential component of the glutamate clearance mechanism, in which glutamate is converted into non-toxic glutamine by glutamine synthetase after transportation to Müller cells [81]. The TES-induced upregulation of glutamine synthetase localized in the Müller cells suggested a neuroprotective mechanism by which TES regulated glutamate metabolism and prevented neuronal injury due to excitotoxicity.

### 4.6. Vasodilation Mechanism

An increased retinal blood flow could explain the therapeutic effects of TES in subjects with retinal ischaemic insult. A clinical study in healthy males demonstrated that a single 30 min session of TES increased the chorioretinal blood flow, which was sustained for 40 h after stimulation without significantly affecting the intraocular pressure and systemic blood circulation [49]. A randomised controlled trial on TES treatments in RP patients showed an improved blood flow in the optic nerve and retina [50]. Although the mechanism of how TES enhanced ocular circulation is unclear, researchers hypothesised that TES induced the release of molecules that mediated vasodilation. The vasodilation hypothesis is strengthened by the finding that TES upregulated IGF-1 expression [34,35], a growth factor that was shown to influence vasodilation through a nitrous oxide synthase-dependent pathway [82,83], which could lead to increased retinal blood flow. Furthermore, TES was reported to induce vasodilation and increase cerebral blood flow in a rat model of cerebral vasospasm through the stimulation of the ophthalmic division of trigeminal nerve endings in the cornea, suggesting the activation of the trigeminovascular reflex [84].

## 5. Potential Antidepressant-like Activities of TES

### 5.1. Brain Regions Stimulated by TES

The application of TES as a treatment for vision restoration was widely investigated. Interestingly, TES was shown to stimulate not only brain regions related to visual processing, but also other unrelated brain regions. A human study utilizing ^18^F-fluorodeoxyglucose positron emission tomography examined brain regions stimulated during TES [29], which showed activation in the occipital cortex, including Brodmann’s Area (BA) 17 in the primary visual cortex, and BA 18 and BA 19 in the secondary visual cortex. There was also activation in the inferior temporal gyrus, which is part of the ventral visual stream involved in visual processing. Aside from the visual cortex activation, enhanced brain activity was also recorded in the bilateral PFC and PHG. Rodent electrophysiological studies found that prolonged TES led to a sustained excitation of the PFC, suggesting that the stimulating effects of TES could diffuse beyond the visual pathway [30,31]. Although the functional implications of increased activity in PFC and PHG by TES remain obscure, they suggest that TES may exert effects on emotional regulation by activating brain regions highly associated with depression. Indeed, antidepressant treatments or psychotherapy were shown to normalize the PFC glucose hypometabolism observed in depressed patients [85,86], suggesting a positive association between PFC activity and the remission of depressive symptoms. A functional magnetic resonance imaging study demonstrated the involvement of the PFC in emotional control, as indicated by an increase in PFC activity during voluntary suppression of negative emotions [87,88]. Similarly, the PHG was shown to have a pathophysiological role in depression, as indicated by the high discriminative power of its functional connectivity in identifying depressed patients from healthy controls [89].

### 5.2. TES-Induced Behavioural Changes in Corneally Kindled Models

The effects of TES on behavioural alterations were previously demonstrated in corneally kindled rodent models [28,90,91]. Corneal kindling is an epileptic model generated through repeated TES at sub-convulsive doses until a generalized seizure is achieved. Wlaź et al. reported that TES induced a significant reduction in despair-like behaviour in fully kindled rats, as demonstrated by a decrease in force swim immobility [28]. Interestingly, such an antidepressant-like response was accompanied by an increase in anxiety-like behaviour in the elevated plus maze test. On the contrary, 6 Hz corneal kindling did not produce an anxiety-like response, instead it resulted in anhedonic-like behaviour in both saccharin preference and novelty suppressed feeding tests [90]. Furthermore, results reported by Wlaź et al. contradicted the findings of another study by Koshal et al., which showed corneally kindled mice had depressive-like behaviour in the tail suspension test [91]. Although TES was shown to induce behavioural changes in kindled models, its prodepressant or antidepressant effects remains unclear due to inconsistent results among studies. More importantly, these investigations were conducted in a fully kindled model, which is not a proper animal model of depression, and could display abnormal behavioural phenotypes that would confound the effects of TES on regulating mood-related behaviour. Moreover, the high stimulation intensities of TES (up to 19 mA amplitude) used to trigger epileptic seizures were inappropriate for examining the antidepressant potential of TES, as such extreme stimulation parameters could lead to tissue damage. Although TES was demonstrated to alter behaviour, its potential therapeutic use in depression needs to be investigated further in a proper animal model of depression using appropriate stimulation parameters.

## 6. Comparison of FDA-Approved Treatments for Major Depression

Interestingly, TES shares several mechanisms of action with some of the existing antidepressant treatments, which can provide a basis for identifying its putative antidepressant properties and the underlying molecular pathways. We compared selective serotonin reuptake inhibitors (SSRIs), repetitive transcranial magnetic stimulation (rTMS), and ECT (Table 1). These depression treatments are approved by the United States Food and Drug Administration (FDA) and are representative therapeutic options for major depression and TRD. Among the different SSRIs options, fluoxetine and escitalopram act to increase serotonin activity by maintaining its extracellular concentration. They are the most commonly prescribed first-line antidepressants with proven safety and efficacy for use in paediatric and adult patients [92,93]. As a non-surgical, non-convulsive brain stimulation therapy, rTMS utilizes a time-varying magnetic field to modulate cortical plasticity and excitability [94,95]. Another non-invasive neuromodulation technique is ECT, which is conducted under general anaesthesia. It intentionally triggers a brief generalized cerebral seizure via the delivery of a small electrical charge to the patient’s scalp [96]. It is speculated that the neurotrophic, anti-apoptotic, and anti-inflammatory activities of TES greatly resemble the biomarker changes observed in the aforementioned antidepressant pharmacological and neuromodulation interventions. 

A growing body of evidence suggests the regulation of neurotrophic signalling has tremendous potential for treating depression. Impaired neuroplasticity is thought to arise from the dysregulated expression of neurotrophins, which have dual roles in regulating neuronal survival and activity-dependent synaptic plasticity [118,119]. Specifically, it has long been speculated that BDNF, a major neurotrophic factor that supports the growth, maturation, and maintenance of nerve cells, plays a direct role in the pathophysiology of depression [120]. A decreased plasma BDNF level was found to be significantly associated with suicidal behaviour in major depression [121]. Similar to TES, the administration of rTMS, ECT, and SSRIs in depressed subjects also led to a neurotrophic enhancement, particularly the enhanced expression of BDNF. Indeed, a 12-week escitalopram treatment in depressed patients reversed the downregulated BDNF mRNA levels in peripheral leukocytes, which normalised serum BDNF levels and alleviated depressive symptoms [97]. Similarly, a small cohort of depressed older patients was treated with escitalopram for 2 months, which resulted in a significant increase in BDNF serum level associated with improvements in the geriatric depression score [98]. Likewise, prolonged rTMS administration in a chronic unpredictable mild stress (CUMS) model of depression upregulated hippocampal BDNF expression for up to 2 weeks after treatment discontinuation and reversed the stress-induced depressive-like behavioural changes [101,102]. Concordantly, several lines of evidence demonstrated that TRD patients who received rTMS showed remarkable increases in peripheral BDNF levels [99,100,103]. Similarly, ECT was shown to upregulate BDNF expression in depressed subjects in both preclinical and clinical studies [104,105,106]. A recent meta-analysis showed that ECT increased peripheral BDNF levels in depressed patients consistent with pharmacological antidepressant interventions, and further highlighted the association between BDNF and the risk of depression [105]. Although the effects of TES on enhancing BDNF levels in the retina are consistently reported [36,38,39,46], its effects on regulating neurotrophin expression in various brain regions involved in emotional regulation, and its associated therapeutic potential, require further investigation.

In major depression, chronic stress can induce excessive apoptotic cell death leading to neurodegeneration in the central nervous system [122]. A balance between pro-apoptotic factors (e.g., Bax and Bak) and anti-apoptotic factors (e.g., Bcl-2 and Bcl-xl) plays a crucial role in controlling the activation of apoptotic pathways [123]. Studies showed that TES could upregulate anti-apoptotic Bcl-2 expression and downregulate pro-apoptotic Bax expression [36,45,46]. Similarly, SSRIs were shown to protect neurons from apoptosis by modulating the expression of the Bcl-2 gene family members that regulate caspase activation and cell death [124]. Fluoxetine treatment in a CUMS model enhanced Bcl-2 expression in the central nucleus of the amygdala, frontal, and cingulate cortices, and reduced Bax expression in the hippocampus, which are all critical brain regions implicated in depression [107,125]. Concordantly, changes in apoptotic markers such as the downregulation of Bcl-2 and upregulation of caspase-3 were remarkably attenuated in CUMS rats after 3 weeks of fluoxetine treatment, further supporting an anti-apoptotic mechanism of SSRIs [108]. Similarly, ECT was found to positively affect Bcl-2 mRNA expression in various sub-regions of the limbic system, while also selectively increasing Bcl-xl mRNA expression in the hippocampus [107]. On the other hand, rTMS was able to repress Bax and augment Bcl-2 expression levels in the hippocampus of CUMS rats, which were accompanied by the amelioration of depression-like behaviour [102,109]. Although TES was also shown to possess anti-apoptotic properties, whether these resulted in antidepressant activity remains an interesting topic for future investigation.

The inflammatory hypothesis of depression is supported by the heightened inflammatory responses found in a significant proportion of the depressed population [126], including increased levels of neuroinflammatory cytokines, which are believed to cause serotonin and melatonin depletion via the neurotoxic kynurenine pathway [127]. Interestingly, changes in the pro- and anti-inflammatory profile of TES greatly paralleled that of rTMS, ECT, and SSRIs. A meta-analysis reported SSRIs to have suppressive effects on inflammatory factors in depressed patients who exhibited increased serum levels of major inflammatory cytokines such as IL-6, IL-1β, and TNF-α [111]. The anti-inflammatory role of SSRIs was further supported by a clinical study on 98 depressed patients which found that treatment with either fluoxetine or escitalopram for 2 months significantly reduced inflammatory markers [110]. Notably, TRD patients treated with rTMS had gradually attenuated serum levels of pro-inflammatory IL-1β and TNF-α accompanied by positive changes in Hamilton Depression Rating Scale-24 scores [103]. Moreover, rTMS exerted antidepressant-like effects in a CUMS model via a nuclear factor-E2-related factor 2 (Nrf2)-dependent anti-inflammatory mechanism, which suppressed the production of pro-inflammatory TNF-α, iNOS, IL-1β, and IL-6 in hippocampal regions [112]. A substantial body of evidence consistently showed that ECT could alleviate pro-inflammatory cytokine secretion in depressed patients, as indicated by a notable reduction in peripheral TNFα, IL-6, eotaxin-3, and IL-5 levels [113,114,116,117]. Moreover, ECT was also reported to increase the level of blood IL-10 [115], which is a well-established anti-inflammatory cytokine that prevents neuronal and glial cell death [128]. Given that TES was demonstrated to reduce inflammatory responses via regulating cytokine expression and suppressing microglial activation [39,42,43,44], its putative anti-depressant effects on inflammation warrant investigation in the future.

## 7. Benefits and Risks of TES As a Depression Treatment

A major advantage of TES is that it is a non-invasive, reversible, and highly adjustable stimulation method. Numerous pre-clinical and clinical studies demonstrated an excellent safety profile of TES and no serious adverse side effects were reported. On the contrary, invasive neuromodulation approaches such as deep brain stimulation carry a risk of infection and haemorrhage during invasive neurosurgery [129,130,131]. Moreover, there is a risk of seizure during stimulation in rTMZ [132], while ECT has potential cognitive side effects including retrograde and anterograde amnesia [133,134]. Compared with conventional antidepressant drugs that affect the entire body, TES can deliver its effects in specifically targeted brain regions [135]. Furthermore, TES avoids the common side effects of antidepressants such as weight change, hepatotoxicity, and gastrointestinal problems [136,137]. The adjustable nature of TES allows the stimulation parameters to be fine-tuned depending on the individual patient’s condition and disease progression. The stimulator can be quickly turned off if an adverse event occurs and the stimulating electrodes can be easily removed. Additionally, the simple administration of TES could allow for self-administration by patients, which greatly enhances its flexibility throughout the treatment period [138].

Although TES is well tolerated in humans with only mild and transient side effects, even after prolonged use, there are some commonly observed side effects, including foreign body sensation, dry eye symptoms, and corneal punctate keratopathy, which can be mostly resolved without further treatment [76,138,139,140]. Other infrequent adverse events include unilateral cataracts, the sensation of flashing lights, muscle twitching, vomiting, and a tingling sensation on the side of the head [138]. Moreover, there were two reported cases of retinal perforation in rats, possibly as a result of mechanical pressure or high charge density stimulation [36]. Considering the potential ocular damage caused by high amplitude stimulation, the optimal stimulation parameters need to be well validated. Furthermore, both antidepressant-like and depressive-like behaviour was reported in the corneal kindling model, and it raised the possibility that TES could resolve or exacerbate depressive-like symptoms. It is therefore important to examine the therapeutic effects of TES using an animal model of depression with proper control, and further investigate of the putative TES-induced activation in brain structures associated with depression in order to delineate the underlying mechanisms of action.

## 8. Conclusions

As an emerging non-invasive neuromodulation approach, TES shows promising neuroprotective properties involving neurotrophic, neuroplasticity, anti-apoptosis, anti-inflammatory, anti-glutamatergic, and vasodilation mechanisms. Although the beneficial effects of TES in visual restoration are widely reported, its putative neuroprotective effects remain largely unexplored. The above studies show that TES can activate brain regions specifically involved in mood alteration. Some studies show that it can induce anti-depressant-like behaviour in corneally kindled models, raising the possibility of using TES in emotional regulation. The hypothesis that ophthalmological treatments can also exert beneficial effects on mood disorders is further supported not only by the interconnections between visual and emotional pathways, but also by the bidirectional association between visual impairment and depression. Here, we speculate that TES may exert antidepressant-like activities similar to some FDA-approved depression treatments including SSRIs, rTMS, and ECT, which are consistently shown to target neurotrophic expression, apoptosis, and inflammatory processes in the nervous system. This review provides evidence of the potential antidepressant effects of TES to encourage further investigations into the therapeutic use of TES in treating depression. Nevertheless, the potential neuroprotective effects of TES need to be properly evaluated in preclinical animal models of depression and in clinical human trials to ensure its safe application and efficacy in depressed populations.

## Figures and Tables

**Figure 1 cells-10-02492-f001:**
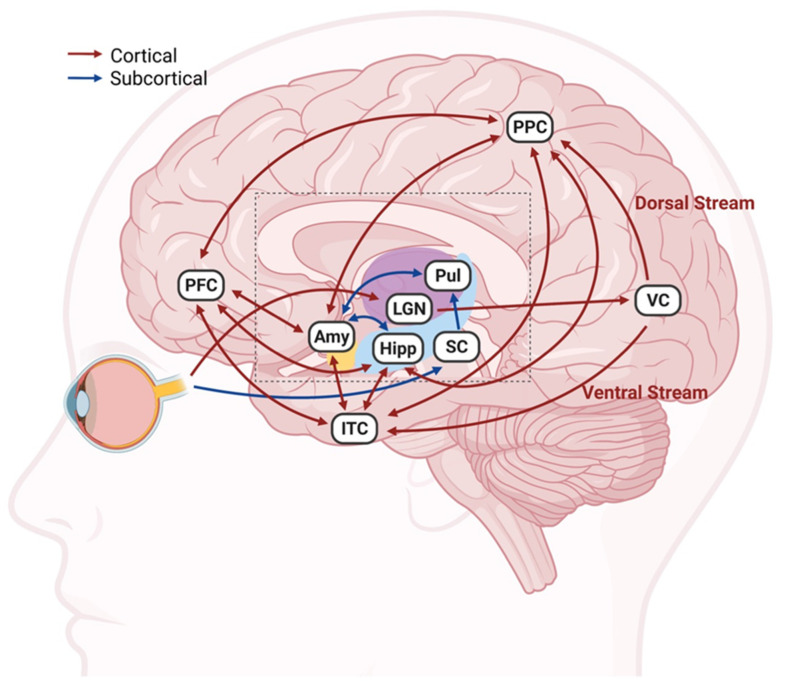
Connections between the visual and emotional systems. The diagram illustrates the major cortical (red arrows) and subcortical (blue arrows) pathways connecting brain regions involved in vision and emotion. In the cortical routes, visual inputs originating from the retina are transmitted to the VC via the thalamic relay station LGN. The VC further projects dorsally to PPC or ventrally to ITC, where they interconnect to the PFC and limbic structures, including Amy and Hipp, which are involved in emotional regulation. Alternatively, visual signals can be conveyed subcortically to Amy via SC and Pul in the thalamus. Amy, Hipp, and PFC can further interchange information directly or indirectly through various relay structures. The less significant intermediate structures and connections in these pathways are not shown to simplify the diagram. Amy: amygdala; Hipp: hippocampus; ITC: inferior temporal cortex; LGN: lateral geniculate nuclei; PFC: prefrontal cortex; PPC: posterior parietal cortex; Pul: pulvinar; SC: superior colliculus; VC: visual cortex. Created with BioRender.com (accessed on 28 August 2021).

**Figure 2 cells-10-02492-f002:**
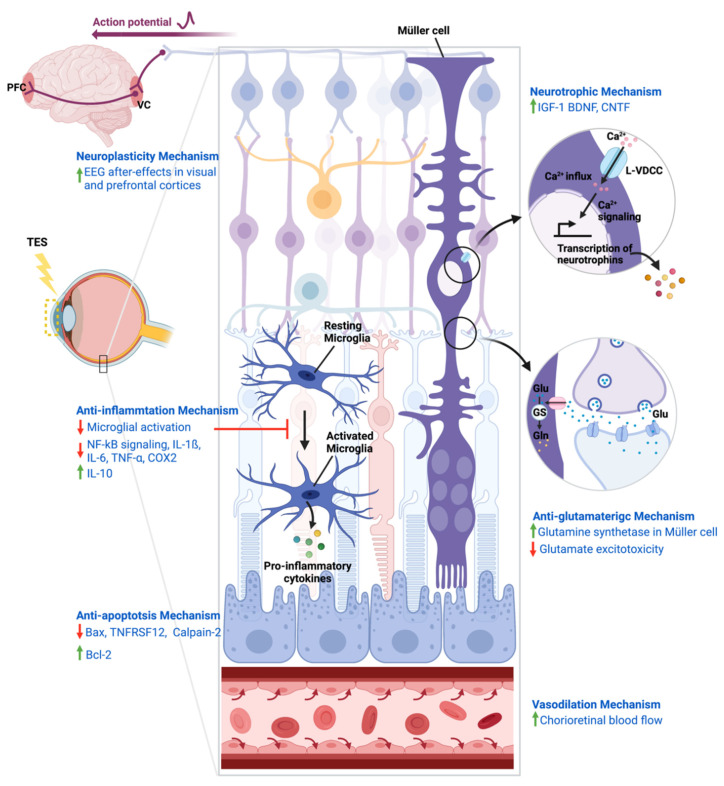
Potential mechanisms of the action of TES. The diagram shows the enlarged structures in retinal layers associated with several proposed mechanisms underlying the neuroprotective effects of TES. Overall, TES increases the survival of neurons and preserves their functions by promoting the expression of neurotrophins in Müller cells, reducing inflammation by inhibiting microglial activation, and preventing apoptotic cell death. TES administration was reported to enhance activities in the VC and PFC, possibly through repeated activation of the visual pathway leading to strengthened synaptic function and neuronal synchronization from the retina to the VC or to the PFC via cortical or subcortical routes. Additionally, TES was reported to rescue cells from glutamate-induced excitotoxicity by enhancing the release of glutamine synthetase from Müller cells. Increasing chorioretinal blood flow through vasodilation is proposed to underlie the protective effects of TES. Bax: Bcl-2-associated X protein; Bcl2: B-cell lymphoma 2; BDNF: brain-derived neurotrophic factor; CNTF: ciliary nerve trophic factor; COX2: cyclooxygenase-2; EEG: electroencephalogram; Glu: glutamate; Gln: glutamine; GS: glutamine synthetase; IGF-1: insulin-like growth factor 1; IL: interleukin; L-VDCC: L-type voltage-dependent calcium channel; NF-kB: nuclear factor κB; PFC: prefrontal cortex; TES: transcorneal electrical stimulation; TNF-α: tumour necrosis factor-α; Tnfrsf12a: TNF receptor superfamily member 12a; VC: visual cortex. Created with BioRender.com (accessed on 28 August 2021).

**Table 1 cells-10-02492-t001:** The shared mechanisms of action of TES for visual improvement and FDA-approved treatments for depression. The associated biomarkers and/or behavioural changes of TES-induced neurotrophic, anti-apoptotic, and anti-inflammatory mechanisms in preclinical ophthalmology studies are compared with those of FDA-approved treatments for depression, namely, the SSRIs, rTMS, and ECT.

Treatment	Study	Subject	Administration Protocol	Results
Neurotrophic Mechanism				
Transcorneal electrical stimulation (TES)	[34]	Male Wistar rats, model of optic nerve transection	100 μA, 20 Hz, 3 ms/phase, 1 h, 1 session	TES increased survival of axotomized RGCs and expression of IGF-1 in Müller cells in the retina.
	[37]	Retinal Müller cell culture	1–10 mA, 20 Hz, 1 ms/phase, 30 min, 1 session	Stimulation increased IGF-1 mRNA and Ca^2+^ reflux which were suppressed by an L-VDCC blocker.
	[38]	Retinal Müller cell culture	10 mA, 20 Hz, 1 ms/phase, 30 min, 1 session	Stimulation increased BDNF mRNA and intracellular protein levels which were suppressed by L-VDCC blocker.
	[36]	Male Sprague Dawley rats, model of light-induced photoreceptor degeneration	Pre-TES: 100–500 μA, 20–100 Hz, 3 ms/phase, 1.5 h, 1 session. Post-TES: 200/300 μA, 20 Hz, 3 ms/phase, 1 session every 3 days for up to 14 days	Post-TES better preserved ONL and retinal functions. TES increased gene and protein expressions of CNTF and BDNF.
	[35]	Male Wistar rats; model of optic nerve crush	100 μA, 20 Hz, 1 ms/phase, 1 h, 1 session on day 0; 2 sessions on day 0 and 7; 4 sessions on day 0, 4, 7, and 10; daily sessions on day 0–12	TES increased IGF-1 immunoreactivity in the retina, and promoted axonal regeneration and survival of RGCs. The axonal regeneration was inhibited with an IGF-1 receptor antagonist.
	[39]	Retinal microglia and Müller cell culture with intense light exposure	300/500/1000/1600 μA, 20 Hz, 3 ms/phase, 1 h, 1 session	Stimulation increased secretion of BDNF and CNTF by Müller cells.
	[46]	C57/BL mice, MNU model of photoreceptor degeneration	100/200 μA, 20 Hz, 3 ms/phase, 3 sessions on days 1, 3, and 6 after MNU injection	TES ameliorated photoreceptor degeneration and increased mRNA levels of BDNF and CNTF.
	[40]	Retinal Müller cell culture	10–500 μA, 10–100 Hz, 0.5–5 ms/phase	TES induced Müller cell proliferation which was blocked by an L-type calcium channel blocker. TES also increased CNTF mRNA level.
Selective serotonin reuptake inhibitors (SSRIs)	[97]	21 patients with depression 23 healthy controls	Escitalopram, 12 weeks	Escitalopram in depressed patients increased leukocyte BDNF mRNA and serum BDNF to similar levels as in controls. Changes in BDNF levels were correlated with symptoms improvement.
	[98]	5 patients with depression 10 healthy controls	Escitalopram, 10 mg/day, 8 weeks	Escitalopram in depressed patients improved GDS scores and increased serum BDNF beyond the levels in controls. The increase in BDNF was correlated to the improved GDS scores.
Repeat transcranial magnetic stimulation (rTMS)	[99]	16 patients with treatment-resistant depression	1 Hz/17 Hz, 5 consecutive sessions with 24 h interval	rTMS improved HDRS scores and increased serum BDNF.
	[100]	26 patients with treatment-resistant depression	Stimulated in the left PFC, intensity 80% MT, 20 Hz, 800 pulses/day, 10 days	rTMS improved HDRS scores and increased plasma BDNF by 23% in responders. A trend of an association between changes in HDRS scores and BDNF levels was found with rTMS.
	[101]	Male Sprague Dawley rats; CUMS model of depression	15 Hz, intensity 100% of device’s maximum power, 60 pulses/train, 15 s train duration, 15 s intertrain interval, 17 trains/day, 1000 pulses/day, 21 consecutive days	rTMS improved depressive-like behaviour and increased BDNF protein levels in the hippocampus.
	[102]	Male Sprague Dawley rats; CUMS model of depression	15 Hz, intensity 100% of device’s maximum power, 15 trains of 60 pulses with 15 s inter-train interval, 900 pulses/day, 7 days	rTMS improved depressive-like behaviour and increased BDNF protein levels in the hippocampus.
	[103]	58 patients with treatment-resistant depression: rTMS (*n* = 19); non-rTMS controls (*n* = 19) 30 healthy individuals: rTMS	Stimulated in left dlPFC, intensity 80% MT, 10 Hz, 1200 pulses of 1 s with 11 s interval, 20 min/session, 5 sessions/week, 4 weeks.	rTMS in depressed patients improved HDRS scores and increased serum BDNF, which was negatively correlated with HDRS scores.
Electroconvulsive therapy (ECT)	[104]	Male Sprague Dawley rats	Stimulated with ear clip electrodes, 100 V, 50 Hz, 1.5 s, duration, 1 session/day, 10 days	ECT increased BDNF protein level in the hippocampus. ECT also decreased BDNF protein level in the VTA, which was necessary for the antidepressant-like effects of ECT.
	[105]	Meta-analysis of 11 studies enrolled patients with unipolar, bipolar and psychotic depression	Stimulated unilaterally or bilaterally on frontal/temporal/frontotemporal positions, 2–3 sessions/week, 6–12 sessions	ECT increased blood BDNF levels.
	[106]	24 patients with depression	Stimulated unilaterally in right frontotemporal and parietal position, maximum charge 1000 mC, 3 sessions/week, 6–13 sessions	ECT improved HDRS scores in responders and remitters, and increased serum BDNF up to 1 month after the last treatment.
Anti-apoptosis Mechanism				
Transcorneal electrical stimulation (TES)	[36]	Male Sprague Dawley rats with light-induced photoreceptor degeneration	Pre-TES: 100–500 μA, 20–100 Hz, 3 ms/phase, 1.5 h, 1 session. Post-TES: 200/300 μA, 20 Hz, 3 ms/phase, 1 session every 3 days for up to 14 days	Post-TES better preserved ONL and retinal functions. TES increased gene and protein expression of Bcl-2 and decreased expression of Bax.
	[45]	Male brown Norway rats	200 μA, 20 Hz, 1 ms/phase, 1 h, one session	TES differentially regulated 490 genes, including downregulation of *Bax and* Tnfrs12a.
	[46]	C57/BL mice; MNU model of photoreceptor degeneration	100/200 μA, 20 Hz, 3 ms/phase, 3 sessions on days 1, 3, and 6 after MNU injection	TES ameliorated photoreceptor degeneration, increased mRNA Bcl-2 mRNA, and decreased Bax and Calpain-2 mRNA levels.
Selective serotonin reuptake inhibitors (SSRIs)	[107]	Male Sprague Dawley rats; CUMS model of depression	Fluoxetine, 5 mg/kg/day, i.p., 21 days	Fluoxetine decreased Bax mRNA in the hippocampus and increased Bcl-2 mRNA in CeA, Cg, and Fr.
	[108]	Male Sprague Dawley rats; CUMS model of depression	Fluoxetine, 3 weeks	Fluoxetine improved depressive-like behaviour, increased Bcl-2 and decreased caspase-3 protein levels in the hippocampus.
Repeat transcranial magnetic stimulation (rTMS)	[102]	Male Sprague Dawley rats; CUMS model of depression	15 Hz, intensity 100% of device’s maximum power, 15 trains of 60 pulses with 15 s inter-train interval, 900 pulses/day, 7 days	rTMS improved depressive-like behaviour, increased Bcl-2, and decreased Bax protein levels in the hippocampus.
	[109]	Male Sprague Dawley rats; CUMS model of depression	10 Hz, intensity 50% of MT, 1 s stimulation duration, 10 s inter-pulse interval, 500 pulses/day, 10 min/day, 5 daily sessions with 2-day intervals for 3 weeks	rTMS improved depressive-like behaviour, decreased Bax mRNA and protein levels in the hippocampus.
Electroconvulsive therapy (ECT)	[107]	Male Sprague Dawley rats; CUMS model of depression	Stimulated with ear clip electrodes, 60 mA, 0.3 s duration, 1 session daily for 21 days	ECT increased Bcl-2 mRNA in Cg, lateral Fr, CeA, and Bcl-xl mRNA in the hippocampus.
Anti-inflammation Mechanism				
Transcorneal electrical stimulation (TES)	[39]	Retinal microglia and Müller cells culture with intense light exposure	300/500/1000/1600 μA, 20 Hz, 3 ms/phase, 1 h, 1 session	Stimulation inhibited microglial activation and microglial secretion of IL-1β and TNF-α.
	[43]	Male Sprague Dawley rats; model of optic nerve transection	200 μA, 20 Hz, 2 ms/phase, 1 h, 2 sessions on day 0 and day 4; or 4 sessions on day 0, 4, 7 and 10 after optic nerve transection	TES increased survival of RGCs, suppressed microglial activation, and reduced TNF-α expression in microglia.
	[42]	Mongolian gerbils; model of acute ocular hypertensive injury	100 μA, 20 Hz, 1 ms/phase, 1 h, 2 sessions on day 1 and 4 for the 1-week experiment; or 2 sessions on day 1 and 4 each week for the 1-month experiment	TES improved retinal function and survival of RGCs, decreased pNF-κB, IL-6, Cox2 and TNF-α expressions, increased IL-10 levels, and suppressed microglial activation.
	[44]	DBA/2 J mice, model of glaucoma	100 μA, 20 Hz, 1 ms/phase, 10 min, every 3 days for 8 weeks	TES improved RGC axonal survival and reduced the number of microglia.
Selective serotonin reuptake inhibitors (SSRIs)	[110]	98 patients with depression	Fluoxetine, 20 mg/day, 8 weeks Or escitalopram, 20 mg/day, 8 weeks	Fluoxetine and escitalopram reduced inflammatory markers, including C-reactive protein concentration, erythrocyte sedimentation rate, and white blood cell count.
	[111]	Meta-analysis of 22 studies on enrolled patients with depression who were taking FDA-approved pharmacological treatments	Different classes of antidepressants were included, primarily SSRIs	SSRIs in depressed patients reduced serum IL-6, IL-1ß, and TNF-α.
Repeat transcranial magnetic stimulation (rTMS)	[112]	Male Sprague Dawley rats; CUMS model of depression	15 Hz, intensity 30% of device’s maximum power, 20 s train duration, 15 min intertrain interval, 900 pulses/day, seven consecutive days	rTMS improved depressive-like behaviour, increased Nrf2 proteins and decreased TNF-a, iNOS, IL-1ß, and IL-6 expression in the hippocampus. These effects were reversed by Nrf2 silencing.
	[103]	58 patients with treatment-resistant depression: rTMS (*n* = 19); non-rTMS controls (*n* = 19) 30 healthy individuals: rTMS	Stimulated in left dlPFC, intensity 80% MT, 10 Hz, 1200 pulses of 1 s with 11 s interval, 20 min/session, 5 sessions/week, 4 weeks	rTMS in depressed patients improved HDRS scores and decreased serum IL-1ß and TNF-a to levels similar to healthy individuals. IL-1ß and TNF-a levels were positively correlated with HAMD scores.
Electroconvulsive therapy (ECT)	[113]	23 patients with depression: ECT (*n* = 15); non-ECT control (*n* = 8) 15 healthy controls	Maximum charge 504 mC, intensity 0.9 A, max duration 7.9 s, 3 sessions/week, 4–18 sessions	ECT in depressed patients decreased plasma levels of TNF-α to levels comparable with healthy controls.
	[114]	15 patients with depression	Stimulated unilaterally on the right d’Elia position, intensity 0.9 A, 10–70 Hz, 0.5–1.0 ms/phase, 6–8 s stimulus duration, max charge 964 mC, 2–3 sessions/week, 12 sessions	ECT decreased serum levels of eotaxin-3 and IL-5 24 h after the last ECT.
	[115]	50 patients with treatment-resistant depression	Stimulated bilaterally in temporal position, square wave pulse, intensity 550–800 mA, 40–90 Hz, 1–2 ms/pulse, 0.5–4 s stimulation duration, maximum charge 1172 mC, 3 sessions/week, 5–12 sessions	ECT increased serum levels of IL-1 and IL-10, and decreased levels of IL-4 and IFN-*γ*.
	[116]	30 patients with depression	Intensity at 1.5 times of seizure threshold, 3 sessions/week, 5–17 sessions	ECT decreased plasma TNF-α level 2 and 4 h after treatment in the 1st, 5th, and last ECT sessions.
	[117]	62 patients with depression	Stimulated unilaterally on the right, 2 sessions/week until patients were asymptomatic/did not improve further for 3 sessions/intolerable side effects occurred	ECT decreased plasma IL-6 level. TNFα level remained unchanged after ECT.

Bax: Bcl-2-associated X protein; Bcl-2: B-cell lymphoma 2; Bcl-xl: B-cell lymphoma-extra large; BDNF: brain-derived neurotrophic factor; CeA: central nucleus of the amygdala; Cg: cingulate cortex; CNTF: ciliary neurotropic factor; Cox-2: cyclooxygenase-2; CUMS: chronic unpredictable mild stress; dlPFC: dorsolateral prefrontal cortex; ERG: electroretinogram; Fr: frontal cortex; GDS: Geriatric Depression Scale; HAMD: Hamilton Depression Rating Scale; IGF-1: insulin-like growth factor 1; IL: interleukin; iNOS: inducible nitric oxide synthase; L-VDCC: L-type voltage-dependent calcium channel; MNU: N-methyl-N-nitrosourea; MT: motion threshold; Nrf2: nuclear factor-E2-related factor 2; ONL: outer nuclear layer; pNF-κB: phosphorylated nuclear factor κB; RGCs: retinal ganglion cells; RP: retinitis pigmentosa; TNF-α: tumour necrosis factor α; *Tnfrs:* tumour necrosis factor receptor superfamily; VTA: ventral tegmental area.

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
