# Peer review of "Neuroprotective Effects and Therapeutic Potential of Transcorneal Electrical Stimulation for Depression"

_cells, 2021, doi:10.3390/cells10092492_

Round 1
Reviewer 1 Report
In this review, the authors observe that the proposed mechanisms of action for the improvement in visual system function via transcorneal electrical stimulation are similar to those cited for the impact of treatments for depression and major depressive disorder. Further pointing out how connections exist between the visual system structures and those that underlie depression, the authors suggest that transcorneal electrical stimulation should be explored as a potential treatment for depression.
The review is well-written and nicely summarizes available literature with regard to the success of TES for the visual system, as well as the various mechanisms of action proposed for retinal ganglion cell survival and improved visual function. The authors also summarize the depression treatments; however, they do not critically review these treatments. Are current treatments for depression somehow deficient such that there is significant need for another approach? This element is missing from the review.
Other items that should be addressed:
lines 60-61: “enhanced activity in prefrontal cortex and parahippocampal gyrus” reference is a StatPearl, not a journal article or original research, and the reference cited does not cover TES-associated activity or these two structures. The authors have other references they can cite here, such as [25, 77 or 78].
Figure 2: In the portion of the schematic showing the brain, the connections from RGCs to VC and PFC are running in the wrong direction. The dots are meant to denote neural cell bodies with the < shape indicating a synapse. The synapse should be from RGCs to VC, not VC to RGC; it should also be from VC to PFC, not PFC to VC.
line 403: Define MDD here (this is its first use in the manuscript)
line 425: Define DBS here, also first use
Section 7: In the context of risks of TES as a depression treatment, the authors should offer a critical appraisal of the literature that connects visual structures and those associated with depression since 1) the connections as described are relatively weak, and 2) there is legitimate controversy about whether kindling-associated corneal stimulation exacerbates depression or resolves it.
Reviewer 2 Report
The review is nicely written with a very good flow of information. The figures are very well made and nicely designed to summarize the information.
I also applaud the efforts of the authors on highlighting the common mechanisms in both the potential treatment of optical diseases and depression with transcorneal electrical stimulation. A couple of minor suggestions:
- Abstract, lines 16-17: replace "neuroplasticity" with "neuroplastic", and "apoptosis" with "apoptotic".
- Introduction, lines 40-41 authors describe well documented associations between depression and other diseases. They should also include pain conditions in their list (i.e. fibromyalgia, post-herpetic neuralgia, chronic pain, etc.).
- Introduction, line 67: add "putative" before "neuroprotective effects" since this is a hypothetical implication.
- Section 3.2 and implication of sentence in Section 4, line 156-157: I am not familiar with what is considered a strong level of evidence in ophthalmology clinical research, but the cited studies in 3.2 seem to be lower level, since they are mostly non-randomized and small sized (low power). This means that the efficacy may not be well-established and perhaps not that "well-studied".
- In Figure 2: define EEG in the list of abbreviations at the end of the caption.
- The title in Table 1 should be more descriptive. This extensive Table contains studies of TES for ophthalmologic animal models and clinical for depression.
